# Adapting the PEERS^®^ for Young Adults Program for Autistic Adults across the Lifespan

**DOI:** 10.3390/healthcare12161586

**Published:** 2024-08-09

**Authors:** Samantha A. Harker, Leslie C. Baxter, Stephen M. Gallegos, Melissa M. Mitchell, Lillian Zerga, Nicole L. Matthews, B. Blair Braden

**Affiliations:** 1School of Life Sciences, Arizona State University, Tempe, AZ 85287, USA; saharker@asu.edu; 2College of Health Solutions, Arizona State University, Tempe, AZ 85287, USA; smgalleg@asu.edu; 3Mayo Clinic, Phoenix, AZ 85054, USA; baxter.leslie@mayo.edu; 4Southwest Autism Research and Resource Center, Phoenix, AZ 85006, USA; mmitchell@autismcenter.org (M.M.M.); lzerga@asu.edu (L.Z.)

**Keywords:** autism spectrum disorder, neurodiversity, adult intervention, participatory, autistic inclusion, social skills

## Abstract

The Program for the Education and Enrichment of Relational Skills (PEERS^®^) is an evidence-based intervention developed for autistic individuals to support social communication, peer interactions, independence, and interpersonal relationships. Despite a demonstrated effectiveness for young autistic individuals in the US and several other countries, PEERS has yet to be modified to support the needs of autistic adults across the lifespan. The present study describes how our team sought autistic voices to adapt PEERS for adults of any age. Specifically, we aimed to address the needs of middle-aged and older adults and adapt the curriculum to be more neurodiversity-affirming. Between two cohorts that completed the program consecutively, we evaluated the acceptability of the adapted PEERS program and made refinements based on feedback from autistic participants and their study partners. Results indicated that Cohort 2 reported higher satisfaction with the PEERS components and overall program than Cohort 1, suggesting effective refinement. We present a framework of adaptations that more specifically address the needs of middle-aged and older adults in a neurodiverse-affirming way compared to previous iterations. Our approach to implementing an adapted PEERS curriculum across the adult lifespan may serve as a model for improved clinical care and cultivate the acceptance of neurodiversity in the interpersonal domains of autistic adults’ lives.

## 1. Introduction

Autistic adults experience persistent social communication challenges beyond young adulthood and even into elderly years. In a meta-analysis (N = 1199), researchers found that many autistic adults struggle to establish and maintain friendships and relationships, which potentially contributes to challenges for leading an independent lifestyle [1]. Autism-related difficulties with social cognition may worsen with age, particularly for men, and may be related to executive functioning challenges [2]. Despite persistent social challenges, many autistic adults desire connection. Specifically, research found that while some autistic adults are engaging socially within communities, it may be hard for them to deepen social relationships beyond basic acquaintances [3]. This suggests that some autistic adults could benefit from support to develop deeper connections and relationships [3] to eventually improve quality of life [4].

Likely related to persistent social challenges, many autistic individuals do not achieve typical adult milestones (e.g., residential independence; competitive employment; advanced education; financial independence/stability; romantic partnership) and have significant adaptive functioning challenges throughout their lifespan [5]. Importantly, autism behavioral characteristics can interfere with adult milestones, day-to-day activities, and co-occurring conditions. These challenges reduce quality of life for aging individuals [4,6] and contribute to the findings that the long-term outcomes for autistic adults are poor [7]. Social and health services are also lacking for autistic adults, resulting in stress for the individual and their family [8]. Taken together, there is a clear need for high-quality outcome research and intervention programs to assist current and future generations of autistic adults [8,9]. However, few autism studies focus on older autistic adults (174 out of 49,793), and most focus on young adults and children [6,10]. Although aging with autism can be challenging, evidence-based services may improve quality of life in the face of lifespan transitions.

Importantly, research has found that loneliness, social isolation, and desire for social relationships were among the most frequently mentioned topics by autistic participants, with autism support groups and services identified as valuable and helpful [11,12]. Participants attested that their lived and shared experiences strongly indicate the need for increased support for reducing isolation and barriers to accessing diagnoses [11]. Unfortunately, autistic voices are underrepresented in the small body of autism aging research, which has contributed to a lack of understanding for autistic adults, their aging experiences, and services [13]. A gap exists in the support available for autistic adults across the lifespan; autistic voices are needed in partnership with researchers to help fill this gap.

### 1.1. Program for the Educational Enrichment of Relational Skills (PEERS^®^)

PEERS is an evidence-based intervention program developed for autistic individuals [14]. PEERS aims to support autistic individuals in gaining skills related to social communication, peer interactions, independence, and interpersonal relationships. PEERS’s feasibility and effectiveness amongst autistic adolescents was first established in 2009, with manualized guidelines available in 2010 [15,16]. In 2012, PEERS was adapted for autistic young adults. Among young adults, efficacy measures indicated improved social responsiveness and empathy [17]. PEERS has been adapted for community settings with adolescents, showing feasibility and promising effectiveness [18]. Others have demonstrated that the long-term benefits of PEERS persist for months [19] and even years [20] post intervention. PEERS has also successfully been adapted for young adults outside North America [20,21,22,23,24,25], those with intellectual disabilities [21], and autistic college students [22]; all demonstrating feasibility, acceptability, and efficacy using various social outcome measures such as social skills, responsiveness, knowledge, cognition, and anxiety, as well as adaptive behavior.

### 1.2. Current Study

Given the strong evidence base indicating program-related gains for autistic adolescents and young adults, we sought autistic voices to assist our team of researchers (including autistic students and consultants) to adapt PEERS for Young Adults to increase the relevance for autistic adults across the lifespan. The overarching goal of this current study is to describe the adaptations we made to the PEERS program with input from autistic adults to increase the appropriateness for autistic adults throughout the lifespan, thereby providing a framework for other areas of research-adapting interventions for autistic adults. Further, we examined the acceptability of the adapted PEERS program between the two cohorts of participants who completed the program in fall 2021 (Cohort 1) and spring 2022 (Cohort 2), since we further refined the program based on feedback from autistic adults and their study partners who participated in Cohort 1. We hypothesized that Cohort 2 would rate the program as more acceptable than Cohort 1.

## 2. Materials and Methods

### 2.1. Study Design

Adult lifespan PEERS was delivered as a part of a pilot randomized controlled trial (RCT; [26]). The study compared adult lifespan PEERS to a newly developed multi-component program, Strengthening Skills. Strengthening Skills included adult lifespan PEERS but with additional complementary components focused on mindfulness-based strategies for managing mental health symptoms and building executive skills. We recruited two cohorts of participants for the RCT to ensure that the program group sizes would be manageable (i.e., to allow adequate time for each participant to contribute to discussions, ask questions, and have the opportunity to practice new strategies). Both intervention groups were compared to a delayed treatment control (DTC) group for preliminary efficacy findings [27]. This paper focuses on acceptability ratings from two consecutive cohorts of participants randomized to the standalone adult lifespan PEERS group. Cohort 1 completed PEERS in fall 2021 (August–December) and Cohort 2 completed PEERS in spring 2022 (January–May).

### 2.2. Participants

The sample included 12 autistic adults (M age = 43.17, SD = 15.08; range: 21–68; 58% male) and 6 program partners (Table 1) who were randomly assigned to the standalone adult lifespan PEERS group. The inclusion criteria for autistic adults were: (1) the self-report of an ASD diagnosis and/or a score of 60 or higher on the SRS-2 adult self-report form at screening; (2) meeting criteria for autism or autism spectrum on the ADOS-2 [23]; and (3) a score of 70 or higher on the KBIT-2 [24]. The majority of autistic adults participating identified as White (n = 10; n = 1 Hispanic; n = 1 other).

### 2.3. Autistic Adult Input

We received feedback from and planned the PEERS adaptation with a rich network of autistic adults and stakeholders. First, four autistic adults participated in focused interviews for us to learn about their persistent challenges and obtain initial feedback on the early stages of planning. This included a 30-year-old male, a 33-year-old female, a 42-year-old female, and a 60-year-old male who all identified as White, with education ranging from college to a completed Master’s degree. These autistic adults identified three personal stakeholders (family or friends) to participate in the focused interviews. Next, we refined the curriculum with a retired autistic occupational therapist and an autistic researcher on the study team. After the study was concluded, we developed an advisory board of four previous autistic participants to guide our dissemination of findings.

### 2.4. Program Delivery

Participants attended 16 weekly 90 min virtual meetings, during which the curriculum was delivered by trained PEERS providers. PEERS themes are displayed in Figure 1 and the weekly lessons in Table 2. Consistent with the telehealth guidelines provided by the UCLA PEERS clinic, the program content was presented on PowerPoint slides using screen sharing, and participants were asked to use gallery mode so that they could see all the other group members. Breakout rooms were used to facilitate in-group practice. A narrative of iterative changes made to the program based on survey responses and the experiences of the facilitators is provided below.

### 2.5. Survey Measures

Participants completed anonymous mid-point acceptability surveys with quantitative open-ended questions about the individual PEERS components delivered during the first half of the program. Quantitative questions prompted participants to rate how helpful each of the program components were to them on a 5-point Likert scale ranging from 1—very unhelpful to 5—very helpful (see Table 3). The open-ended questions were as follows: (1) what are the program’s strengths?; (2) what are the program’s limitations?; (3) what recommendations do you have for addressing these limitations or improving the program in other ways?; and (4) is there anything else you would like us to know?. Final acceptability surveys included Likert scale questions regarding the helpfulness of individual PEERS components across the entire program (see Table 3) as well as the same open-ended questions about the entire program’s strengths, limitations, and recommendations for addressing limitations. Additionally, participants completed a 17-item acceptability survey adapted from Stahmer and colleagues [25] using a Likert scale ranging from 1—strongly disagree to 7—strongly agree (see Table 4). Final surveys were not anonymous because we planned to examine associations between participant demographic characteristics and the acceptability ratings. As the sample was too small for meaningful statistical analyses and numerical means, the variance in quantitative values is described. Two Cohort 2 participants completed the mid-point survey but did not complete the final survey.

## 3. Results

### 3.1. Adaptations to PEERS^®^ for Young Adults

#### 3.1.1. Adaptation Theme—Avoiding Perceived Ableism

Prior to the RCT, we adapted the PEERS^®^ for Young Adults protocol based on feedback from focused interviews with autistic adults [26]. Adaptations were made with consultation from an autistic occupational therapist and in collaboration with an autistic doctoral student that was a member of the research team. First, we adapted the social coaching component of the PEERS for Young Adults program, where the social coach is typically the autistic young adults’ parent or guardian. In adult lifespan PEERS, autistic adults were able to choose any person in their lives (autistic or non-autistic) to participate with them, and they were termed the “program partner” to minimize perceived ableism towards the autistic adults. To reduce potential power dynamics between autistic adults and their program partners, program partners were asked to complete the same weekly homework assignments as autistic adults (e.g., joining a new social group; making in-group phone calls). We also introduced the term “collaborative learning” to replace the term “social coaching”, which involved autistic adults and program partners checking in with each other outside of group meetings and discussing progress on weekly homework assignments. For the first and last week’s meeting, all participants met in the same virtual room to allow for introductions, and to emphasize that the program partners were working on their own social goals during the program. Autistic adults and the program partners met in separate groups for weeks 2 through 15 to maintain consistency with the original program.

We initially planned to require autistic adults to participate with a program partner (e.g., family member; friend). We removed this requirement during recruitment for the larger study after realizing that many autistic adults did not have someone in their life able or willing to participate with them. In PEERS for Young Adults, a social coach (e.g., parent or guardian) is required to participate. In the recruitment phase of our study, we found that many middle-aged and older autistic adults did not have a person who was able to complete the program with them. Therefore, requiring a program partner in adult lifespan PEERS would severely limit accessibility to the autistic adults who may need it most. In our adaptation, a program partner was not required but rather was encouraged. Lastly, given that approximately half of the participants were in romantic relationships, we encouraged these participants to share their own experiences during group conversation about the dating curriculum.

#### 3.1.2. Adaptation Theme—Increasing Relevance to All Ages

Changes resulting from feedback after the mid-point survey were aimed at increasing the relevance of the program to autistic adults of all ages. For example, the role play videos recommended for use when implementing the program through telehealth are set in what appears to be a library and include young adult actors. Additionally, some include topics that may not be relevant for adults past young adulthood (e.g., teasing someone about hanging out with their parents). Although some participants identified the role plays as a strength of the program because they demonstrated program strategies, other participants indicated that the limitations of the role play videos made the program feel like it was developed for a younger demographic. Thus, we wrote new role plays that could be acted out by the PEERS group leader and assistants during the Zoom meeting. Although the new role plays focused on the same social strategies, the dialog was changed to be more reflective of social situations that may be experienced by adults of all ages. Example role plays are reported in Table 5.

In PEERS for Young Adults, there is a strong focus on making new friends, which includes the homework assignment to identify and join a new social group related to one of the participant’s interests. Some participants were primarily interested in improving their existing social relationships; thus, making new friends was not one of their social goals. Many of these participants also had demanding work and/or family responsibilities and they reported that this homework assignment caused a significant amount of undue stress. Based on this feedback from Cohort 1, joining a new social group was made optional but encouraged if the participant had a goal of making new friends. Additionally, participants were encouraged to think about social groups related to their interests that they could join in the future if their goals regarding new friendships changed. Another component of the program that became more flexible for Cohort 2 was practicing strategies through behavioral rehearsals. Emphasizing choice and alignment with individualized goals, participants in Cohort 2 were able to choose whether they wanted to behaviorally rehearse a strategy or not.

Based on feedback from both cohorts and our Community Advisory Board, we added discussions to the weekly meetings about circumstances in which autistic adults may (or may not) choose to mask or disclose their diagnosis. The group leader facilitated the initial discussion with two broad discussion questions (i.e., are there social relationships that you have or would like to have where masking would be helpful/harmful? Are there social relationships that you have or would like to have where you think disclosing your diagnosis (or sharing autistic characteristics about yourself without labeling) would be helpful/harmful?). Participation in the discussion was voluntary, and the group leader emphasized that participants would likely have different experiences and opinions. The group leader did not offer advice or suggestions; instead, they emphasized that deciding when to mask and/or disclose was a personal decision. After the initial discussion, the group leader referred back to the concepts of masking and disclosure when introducing new PEERS strategies. We piloted this approach with the delayed treatment control group of the participants who were also a part of this study. This component of the program was identified as important to our autistic participants who tended to be older and navigating independent work and living environments but seemed to be less critical to participants representing the younger demographic for which PEERS for Young Adults was originally created.

#### 3.1.3. Adaptation Theme—Supporting Neurodiversity

A final adaptation theme included modifying program language to promote neurodiversity and minimize perceptions that the program was encouraging masking. The following adaptations were made based on Cohort 1’s final survey feedback. Instead of describing the curriculum as teaching “social skills”, we presented it as a method for “deconstructing social interactions” to better emphasize agency. This included discussions regarding adults’ agency to choose when and how to implement any strategies they learned in PEERS. We emphasized that PEERS strategies can reduce risk in social situations but there are rarely “right” or “wrong” social choices. We frequently reminded participants that the purpose of the adapted program was to equip them with more or alternative social strategies so that they were more likely to achieve the social outcomes that they desired.

We also modified the language used to introduce and discuss role plays. The original PEERS for Young Adults manual recommends introducing role plays by telling participants to “Watch this and tell me what I am doing wrong (or right)”, which is language that was retained from the initial PEERS for Adolescents program. This language, as well as the overall approach of using concrete rules and steps, was developed to align with the concrete and rule-based cognitive style of some autistic individuals [28]. In the current study, we received participant feedback that this language suggests that ASD behavior is wrong or weird. To address this limitation, we changed the language to “Tell me what I am doing that could be risky” and “Tell me which social strategy I am using”. We also changed the responses to the perspective-taking questions presented after each role play. Instead of focusing on the traits (e.g., weird; strange) of the protagonist who was engaging in “risky” social behavior, the responses reflected confusion or discomfort of the role play partner. Consistent with changes to the role play language, we reworded “social rule” to “social strategy you could try” and changed the wording of PEERS strategies that began with “Don’t” to “Avoid” (e.g., avoid being an interviewer). Lastly, as an extension of our initial changes to the dating curriculum, we encouraged participants to share alternative social strategies in any context that have or have not worked for them in the past. We aimed to capitalize on the powerful model of autistic adults learning from each other. These changes reflect a neurodiversity-affirming approach by removing a perceived association between autism and “wrong” behavior. They also more appropriately respect the autonomy of autistic adults, who have more life experience to draw on than autistic youth.

### 3.2. Survey Results

Generally, participants rated the adapted PEERS program components as helpful (Table 3). Some of the highest rated components across both cohorts were related to conversations and also handling disagreements, which were rated near “Very Helpful” (Table 3). Some of the less popular topics were electronic communication, dating strategies, and handling teasing, which were rated between neutral and helpful (Table 3). This is not surprising given the older age of our participants compared to the initial target demographic of these lessons being autistic adolescents. Cohort 2 rated seven out of nine components higher than Cohort 1 (Table 3), suggesting that the adaptations made to the program based on Cohort 1’s feedback may have made it more acceptable. In particular, Cohort 2 rated behavioral rehearsals an average of one point higher than Cohort 1, which likely reflects the added flexibility in choosing which strategies they would like to rehearse.

Participants rated statements about the adapted program as a whole positively. The mean ratings across Cohort 1 and 2 showed that on average participants ranged from “Slightly Agree” to “Strongly Agree” with each positive statement (Table 4). Some of the strongest endorsed statements reflected that the lessons and homework assignments were clear, that they felt supported in learning new strategies, and that the group leaders were knowledgeable, which were all rated between “Agree” and “Strongly Agree” (Table 4). Cohort 2 rated the program higher on most statements than Cohort 1. Specifically, Cohort 2 rated that the program was more enjoyable, effective in teaching new strategies to improve social relationships, and that the goals were important to their functioning at home/community than Cohort 1, with the averages an entire point closer to “Strongly Agree”. Cohort 2’s higher ratings about statements regarding the program also suggest that adaptations based on Cohort 1’s feedback improved acceptability.

## 4. Discussion

Despite a significant need for social communication support programs for autistic adults throughout the lifespan, this is the first study to our knowledge to systematically refine an existing group-based intervention to increase appropriateness for autistic adults of all ages. Here, we introduce adaptations to the widely effective PEERS for Young Adults program to develop a social communication support program that is suitable for autistic adults across the lifespan. Changes were made to both the content and andragogical approaches of PEERS to increase relevance and acceptability for autistic adults in midlife and older age brackets. Programmatic approaches were adapted and derived from participants and autistic professionals’ feedback, which included avoiding perceived ableism and inequitable power dynamics, increasing relevance to all ages, and supporting neurodiversity. Our findings, using these adaptions, show that Cohort 2 reported higher satisfaction with the PEERS components and overall program than Cohort 1 and indicate that the curriculum and approach were effectively improved. This suggests that these modifications may allow practitioners to serve autistic adults across the lifespan using PEERS strategies while cultivating the acceptance of neurodiversity in the interpersonal domains of autistic adults’ lives.

Previous studies of PEERS and published manuals for the program primarily focused on adolescents and young adults [14,17]. The program was designed using concrete rules, steps, and language to align with the preference of many autistic individuals for rules and routine [28]. This rules-based approach to learning new material aligns well with the developmental periods of adolescence and young adulthood, during which the abstract concepts and “gray areas” of social communication may be difficult to teach in a short-term intervention. However, in alignment with increased expectations for independence during adulthood, a rules-based curriculum may be less effective, with reduced authenticity for adults. The adaptations described in this paper add relevance to the program by emphasizing the agency of autistic adults throughout the lifespan while continuing to use foundational PEERS content to facilitate an improved understanding of social relationships [15,26].

For older autistic adults, the family member(s’) role as a “social coach” in PEERS is less likely to be helpful than it is for adolescents and young adults. For older autistic adults experiencing or seeking greater autonomy in their lives, having a close family member (e.g., parent, spouse, sibling) identified as a “coach” may seem inauthentic, promote unhealthy power dynamics, and works against the goal of developing independent skills in adults. Instead, the integration of the “program partner” approach, who is expected to work on their own social relationship goals and engage in collaborative learning with their partner, allows for the acknowledgement of age-appropriate autonomy. We also emphasized choice/independence instead of a rules-based approach by describing behaviors as “risky”, but always emphasizing the ability to choose.

Other changes made to PEERS included addressing the needs more specifically experienced by middle-aged and older autistic adults. Specifically, the program recognized that the needs of working adults who have achieved some (if not full) independence requires consideration of how social behaviors and the expectations of others fit into their lifestyle, beliefs, and relationships. Thus, it is necessary to include a discussion about when one might disclose their diagnosis and when one would not, whether altering behavior is masking, and knowledge of why certain behaviors may elicit certain reactions from neurotypical individuals. The implementation of these discussions, in alignment with the literature, is deemed critical for middle-aged and older autistic adults, noting that social support is the hallmark of a positive quality of life [27]. Middle-aged and older autistic adults’ success is dependent upon navigating issues at the intersection of the individual and their environment [28], and incorporating the discussion of important topics such as disclosure is an important aspect to any intervention aimed at supporting and validating autistic adults.

Many of our adaptations aimed to further support neurodiversity. Neurodiversity-affirming care has been demonstrated to be a highly positive and impactful for autistic adults [29]. Despite this, most programs have little or no input from autistic individuals, their families, and their communities to strengthen long-lasting and impactful positive support strategies [29]. Beyond simply leveraging positive psychology in neurodiversity-affirming care, our adaptions encouraged a participatory strengths-based approach, participant feedback, and ultimately created a safe environment for autistic adults to learn from one another [30]. This model may be effective to leverage PEERS for improved quality of life for the vast spectrum of autistic adults across the lifespan [31].

To maintain consistency with the original program, autistic adults and program partners met in separate rooms during weeks 2 through 15. This may be viewed as a limitation considering the goal to reduce power dynamics between autistic adults and their program partners. However, a discussion with our autistic Community Advisory Board affirmed that participants valued this separation because it created what was viewed as a safer environment, where autistic adults could feel comfortable sharing with and learning from each other without potential influence from their program partners. Future research that examines the potential benefits of a fully integrated program is needed to determine the drawbacks or strengths of both approaches.

### 4.1. Future Suggestions for PEERS across the Lifespan

To improve the effectiveness of the PEERS across the lifespan program, several suggestions can be implemented for future studies. First, a sampling strategy that integrates participants of diverse age-related, socioeconomic, ethnic, and cultural backgrounds may broaden and benefit the feasibility of the program. Including additional surveys and modalities for feedback from all vested parties, autistic participants, family members, and care providers will be helpful for a close evaluation of the effective aspects and those that need improvement. A long-term follow up should be used to ensure the usability of skills taught in PEERS for real-world applications in a broad range of locations and participants. Lastly, accessible supplementary materials, such as written examples of scenarios involving PEERS topics and general social strategy suggestions, should be developed and tailored for participants and their support system so they can deploy learned skills and embrace neurodiversity.

### 4.2. Limitations

Since the sample size was small and inadequate for comparing satisfaction between cohorts, larger studies are needed to replicate the current findings and to examine the effectiveness of adapted PEERS for adults across the lifespan. Additionally, not all participants had a program partner, which may have affected autistic adults’ perceptions of the benefits of the program. Larger evaluations of the modified program should evaluate the influence of a program partner on autistic adults’ satisfaction with the program. Finally, similar to the originally developed PEERS program, our adapted PEERS program did not include the participation of autistic adults with an intellectual disability, so this approach does not generalize to the full spectrum of autistic adults. Future studies should examine approaches that may address the needs of autistic adults with and without intellectual disabilities.

## 5. Conclusions

Based on the satisfaction survey scores and the increasing improvements in ratings across program iterations, the adaptions we made to PEERS for Young Adults to increase the relevance and acceptability of the program for autistic adults across the lifespan improved program satisfaction. This suggests that most PEERS strategies are valuable for many midlife and older adults, but may be more accessible when presented with stage-of-life modifications of the teaching style, language, choice, and group discussion. In particular, adaptations like those described here may increase the relevance and buy-in of the program to midlife and older adults. Our work to develop the modified version of PEERS may provide the resources and support necessary for other future interventions by empowering and supporting autistic adults’ neurodiversity through interpersonal skills, independence, and the quality and enjoyment of socialization across the lifespan.

## Figures and Tables

**Figure 1 healthcare-12-01586-f001:**
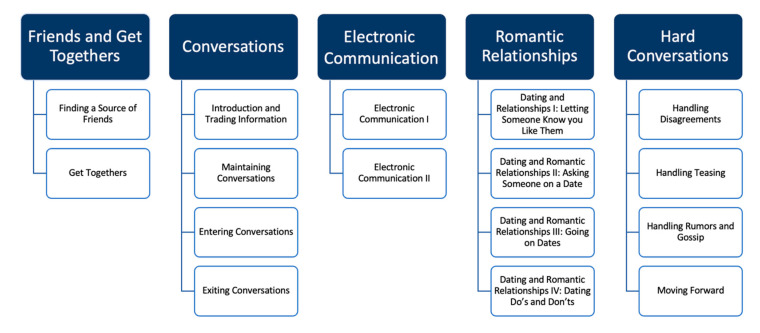
Schematic showing the PEERS^®^ topics and their components.

**Table 1 healthcare-12-01586-t001:** Descriptive statistics by group.

	Cohort 1 (Fall 2021)	Cohort 2 (Spring 2022)	Cohorts 1 and 2
N	10	7	17
Autistic Adult	7 (70%)	5 (71%)	12 (71%)
Age Mean (±SD); range	39.7 (9.35); 25–55	45.86 (21.52); 21–68	42.24 (15.25); 21–68
Gender (Male(%)/Female(%))	(5 (50%)/5 (50%))	(4 (57%)/3 (43%))	(9 (53%)/8 (47%))

**Table 2 healthcare-12-01586-t002:** PEERS^®^ content by week.

Week	PEERS^®^ Lessons
1	Introduction and Trading Information
2	Maintaining Conversations
3	Finding a Source of Friends
4	Electronic Communication I
5	Electronic Communication II
6	Entering Conversations
7	Exiting Conversations
8	Get-Togethers
9	Dating and Romantic Relationships I: Letting Someone Know You Like Them
10	Dating and Romantic Relationships II: Asking Someone on a Date
11	Dating and Romantic Relationships III: Going on Dates
12	Dating and Romantic Relationships IV: Dating Do’s and Don’ts
13	Handling Disagreements
14	Handling Teasing
15	Handling Rumors and Gossip
16	Moving Forward

**Table 3 healthcare-12-01586-t003:** PEERS Components ratings by group. PEERS components were evaluated by the following ratings: (1) very unhelpful, (2) unhelpful, (3) neutral, (4) helpful, and (5) very helpful.

	Cohort 1 (Fall 2021)	Cohort 2 (Spring 2022)	Cohorts 1 and 2
	Mean (±SD)	Mean (±SD)	Mean (±SD)
Two-Way Conversations	4.5 (0.71)	4.43 (0.53)	4.47 (0.62)
Electronic Communication	3.6 (0.7)	4.29 (0.49)	3.88 (0.7)
Humor Feedback	4.1 (0.57)	4.29 (0.76)	4.18 (0.64)
Joining Group Conversations	4.4 (0.7)	4.71 (0.49)	4.53 (0.62)
Exiting Group Conversations	4.5 (0.85)	4.71 (0.49)	4.59 (0.71)
Source of Friends	4.3 (1.06)	4.57 (0.53)	4.41 (0.87)
Get Togethers	4 (0.82)	4.43 (0.53)	4.18 (0.73)
Dating Strategies	3.6 (0.7)	3.71 (0.76)	3.65 (0.7)
Handling Disagreements	4.8 (0.42)	4.17 (0.98)	4.56 (0.73)
Handling Teasing	3.7 (0.82)	4 (0.58)	3.82 (0.73)
Rumors and Gossip	3.9 (0.74)	3.86 (0.69)	3.88 (0.7)
Role Plays	3.3 (0.82)	3.86 (0.69)	3.53 (0.8)
Behavioral Rehearsals	3 (1.15)	4 (1)	3.41 (1.18)

**Table 4 healthcare-12-01586-t004:** Acceptability ratings by group. The PEERS program was evaluated by the following ratings: (1) strongly disagree, (2) disagree, (3) slightly disagree, (4) neutral, (5) slightly agree, (6) agree, and (7) strongly agree.

	Cohort 1 (Fall 2021)	Cohort 2 (Spring 2022)	Cohort 1 and 2
	Mean (±SD)	Mean (±SD)	Mean (±SD)
This is an acceptable program for learning strategies to improve my social relationships.	5.8 (1.14)	6.29 (0.76)	6 (1)
I enjoyed this program.	5.8 (1.14)	6.71 (0.49)	6.18 (1.01)
I would suggest this program to other autistic adults.	5.8 (1.03)	6.43 (0.53)	6.06 (0.9)
The lessons were clear and understandable.	6.3 (0.82)	6.43 (0.53)	6.35 (0.7)
The amount of support I received was sufficient for me to learn the program strategies.	6.5 (0.71)	6.29 (0.49)	6.41 (0.62)
The group leaders were knowledgeable.	6.1 (0.99)	6.86 (0.38)	6.41 (0.87)
The goals of the program were important to my functioning at home and in the community.	5.2 (1.75)	6.29 (0.49)	5.65 (1.46)
I use the strategies I learned during this program in my daily life.	5.5 (1.08)	5.71 (1.25)	5.59 (1.12)
I understand how to use the techniques I learned in my daily life.	6.1 (0.88)	6.14 (0.69)	6.12 (0.78)
I understand which strategies I am working on and why.	5.7 (0.95)	5.29 (1.8)	5.53 (1.33)
The homework assignments were clear.	6.4 (0.52)	6.57 (0.53)	6.47 (0.51)
The homework assignments were manageable.	6.1 (1.2)	5.86 (1.35)	6 (1.22)
The program was effective in teaching me new strategies to improve my social relationships.	5.5 (1.08)	6.43 (0.53)	5.88 (0.99)
I feel that I improved my social relationships as a result of this program.	5.6 (1.07)	6.29 (1.11)	5.88 (1.11)
The program will produce lasting improvement in my social relationships.	5.5 (1.18)	6.29 (0.76)	5.82 (1.07)
I use the strategies I learned from this program regularly.	5.5 (0.85)	6 (1.41)	5.71 (1.1)
I will continue to use the strategies I learned after the program is over.	5.9 (0.99)	6.29 (0.76)	6.06 (0.9)

Note. Survey items were adapted from [25].

**Table 5 healthcare-12-01586-t005:** Examples of revised role plays.

Topic	Risky Role Play	Role Play Demonstrating PEERS Strategies
Sharing the Conversation	**Leader:** “Hey, [Assistant Name], what have you been up to?” **Assistant:** “Oh, not much, just kind of slammed with work, school, life. You know how it goes.” **Leader:** “Oh yeah, I do, you wouldn’t believe how busy I’ve been lately!” **Assistant:** “Wow, what’s been going on-” **Leader *(interrupts)*:** “Between all my responsibilities at work, all the stuff I’ve been doing at home, and my hobbies, it’s like I’m constantly on the go from one thing to the next!” **Assistant**: “Dang, that sounds like a lot-” **Leader *(interrupts again)*:** “Yeah, it really is. Sometimes people wonder how I’m able to keep up with it all. Everyone at work knows things would absolutely fall apart without me, and don’t get me started on everything around the house. I get home, have to put all my stuff away, feed my pets, get started on dinner, then I’ve got to clean up all the dishes after that” **Assistant: *(looks bored, disinterested)*** **Leader *(continues)*:** “and then by the time THAT’S done, it’s like, what next, do I put a load of laundry in, or spend time on my hobbies? It’s good that I’m so skilled at managing my time, because otherwise none of this would get done. It’s just like, when do I get time just for myself, you know?” **Assistant: *(does not respond, attention has drifted away, looking at their phone)***	**Leader:** “Hey, [Assistant Name], what have you been up to?” **Assistant:** “Oh, not much, just kind of slammed with work, school, life. You know how it goes.” **Leader:** “Yeah, I hear you on that. Have you found the time to do anything fun recently?” **Assistant:** “Yeah, actually. You won’t believe this, but I just started taking a robotics class. I know, it’s a little silly.” **Leader:** “Silly? No way, that sounds awesome! I’ve always wanted to do that.” **Assistant (brightens):** “Oh, really? It’s been really fun so far. I’m definitely learning a lot and getting better at it. We programmed a new robot last week and it was amazing!” **Leader:** “Oh I bet, I’m so intrigued by coding, but I’ve never really tried coding on my own. It seems like it would be tricky. What do you think you’ll do in class this week?” **Assistant:** “I’m not sure yet, but it’s always something good. The next time enrollment is open, I’ll shoot you a text, it might be fun for us to do it together.” **Leader:** “That would be great, maybe we could even carpool, if you want?” **Assistant:** “Yes, that’s perfect. I’m excited!”
Joining Group Conversations	**Assistant 1:** “So then I said, ‘No, I think YOU need to take a closer look at my email, Janet.’” **Assistant 2 (chuckles):** “It’s always something around here, I swear.” **Assistant 1 (rolls eyes):** “I know, right? I try to be flexible, but sometimes I’m just like…hello? Am I the only one who knows what is going on?” **Assistant 2 (nods):** “Seriously, you KNOW I can relate to that. Do you remember the time I was on the phone and-“ **Leader (interrupts):** “How about that comedy show, huh? I can’t believe it hasn’t sold out yet!” **Assistants 1 and 2 (collectively turn their heads with confused expressions)** **Assistant 2:** “…what? Anyway [Assistant 1 name], I was on the phone and then Rebecca came over and...” **Leader (interrupts again):** “It’s at the comedy club downtown and while I don’t want to tell you what to do, I HIGHLY recommend getting tickets ASAP because it WILL be a sell-out show!” **Assistant 2 (looking even more exasperated):** “…okay?” **(rolls eyes, annoyed)** “At this point, it’s not even worth telling the story, but…you know what I mean.” **Assistant 1 (glances towards Leader and glances back at Assistant 2 meaningfully):** “I DO know what you mean. We’ve got to go out sometime and just let loose and forget about all this work stuff.” **Leader:** “Sounds like this show would be the perfect opportunity for that!” **Assistants 1 and 2 (make eye contact, no response)**	**Assistant 1:** “…So then I said, ‘No, I think YOU need to take a closer look at my email, Janet.’” **Assistant 2 (chuckles):** “It’s always something around here, I swear.” **Assistant 1 (rolls eyes):** “I know, right? I try to be flexible, but sometimes I’m just like…hello? Am I the only one who knows what is going on?” **Assistant 2 (nods):** “Seriously, you KNOW I can relate to that. Do you remember the time I was on the phone and Rebecca came up and totally interrupted me? Right in the middle of the call? I guess it’s just one of those hazards of having coworkers.” **Assistant 1:** “We’ve got to go out sometime and just let loose and forget about all this work stuff.” **Leader: (looks up at Assistants, smiles, and looks away)** **Assistant 2:** “Yeah, I agree. I was actually looking at a local event calendar the other day, but I didn’t see anything amazing.” **Leader:** “Hey I just overheard you talking about local events. I actually heard about an event this weekend from a friend, it sounds fun, kind of different.” **Assistant 2 (neutral):** “Oh, really? **Leader:** “Yeah, apparently there’s a comedy club downtown and they’re doing something like an improv/open mic night.” **Assistant 1 (opens the circle):** “Comedy? Really? I didn’t know there was a comedy club downtown!” **Assistant 2:** “I’m not usually into comedy, but it also sounds like it could be pretty funny.” **Leader (nods):** “That’s what I was thinking. **Assistant 2 (looks over at Assistant 1):** “What do you think? Up for something a little different?” **Assistant 1 (shrugs and smiles):** “Sure, I’d be down to try it out. Thanks for the suggestion!” **Leader (smiles back casually):** “Yeah, no problem. (pauses) I’m [Leader name], by the way.” **Assistant 2**: “Thanks [Leader Name], I’m [Assistant 2 Name], and this is [Assistant 1 Name].”

## Data Availability

Data available upon request.

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
