# Peer review of "Adapting the PEERS® for Young Adults Program for Autistic Adults across the Lifespan"

_healthcare, 2024, doi:10.3390/healthcare12161586_

Round 1
Reviewer 1 Report
Comments and Suggestions for Authors
Attached I present the results of the review of the paper titled: "Adapting the PEERS for Young Adults Program for Autistic Adults Across the Lifespan".
This is a very interesting and well-written work. But in my opinion there is a lot of room for improvement.
Keywords:
All keywords are included in the title. Authors should think about whether other similar names not included in the title would make indexed searches more efficient.
Introduction:
The introduction does its job well. With clear and concise language, it presents scientific evidence necessary for the paper. However, I think the authors should improve some details:
It would be advisable to introduce the general guidelines of the Peers program in a schematic way. It is indicated that it is a program to improve certain skills, and from there evidence of results is presented. I think it would be interesting to talk a little more about PEERS.
Although it is convenient to present the general objectives of the work in the introduction, the authors nevertheless begin by introducing aspects that do not correspond to this section and that should be located in the method. From line 80 onwards the authors tell us about methods, samples, etc... That goes beyond what the introduction should be and should be included in the method.
2. Materials and Methods
In general the section is very well detailed. However, it would improve by including some important elements:
Nowhere is the rest of the people participating in the study without being part of its sample clearly discussed and explained. It is sensed in the context of work. But it should be clearer.
Could a summary table of the distribution of work during the 16 sessions be included?
Many of the qualitative elements are presented in this section, when they are really "qualitative results"
Results.
Shouldn't qualitative results appear in the results section?
Taking into account the small number of the sample for quantitative analyses, are the previous assumptions taken into account when using the T test? If so, why don't they show up? If not, wouldn't it be better to use non-parametric tests?
Line 266.- "and the difference approached statistical significance" so, is not sigfinicative....
I think that the authors waste in the results section the conclusions that can be reached qualitatively, especially when due to the small number of the sample it is possible that the quantitative results can't help corroborate them through other procedures, taking into account that the sample is very small.
I think they should rethink the paper, because I honestly think what they have done is wonderful work, and it would be convenient for them to improve it to make it known to the scientific community.
Reviewer 2 Report
Comments and Suggestions for Authors
This study focuses on adapting the PEERS program for autistic adults throughout their lifespan, with particular emphasis on how the program developed for young adults can be tailored for middle-aged and older adults. The primary objectives of the study include reaching a broader participant base, conducting more detailed evaluations of feedback, monitoring the long-term effectiveness of the intervention, and examining how the program operates in different cultural contexts. Additionally, supplementary factors such as parental or caregiver feedback and support materials are addressed to enhance the program's success. The findings of this study indicate significant progress in making the PEERS program more suitable for autistic adults across their lifespan and provide a roadmap for future research endeavors.
To broaden the scope of your study, consider adopting a sampling strategy that includes participants from different age groups and socioeconomic backgrounds.
When gathering feedback, carefully evaluate which aspects are more effective and identify areas where further improvements could be made in more detail.
Conduct long-term follow-up studies to assess the effectiveness of the adaptation in real-world conditions.
In evaluating the effectiveness of the program, take into account feedback not only from participants but also from parents or caregivers.
Provide additional resources and support materials tailored to participants and their families to facilitate the implementation of the program.
Disseminate research findings through various communication channels to reach a wider audience.
Offer recommendations for future research based on this study, such as longitudinal monitoring of adaptation effectiveness or exploring how the program operates in different cultural contexts.
Comments on the Quality of English LanguageMinor editing of English language required
Reviewer 3 Report
Comments and Suggestions for Authors
Dear authors, thank you for the opportunity to read your work.
It is a job that has been of interest to me because of its relationship with my work activity.
I think it presents methodological problems that should be corrected.
The topic that the authors discuss is of great relevance and interest. The information (data) on which the research is based is relevant. I believe that their treatment of this information (quantitative and qualitative analysis) is incomplete and requires significant improvement. The results section is evidence of the need for improvement in the analysis processes. The discussion and conclusions presented are not supported by the results.
The modifications that the work requires are so important that it is recommended to redo the method, results and discussion sections.
[Author] General objective: Describe the adaptations made in the PEERS program Secondary objective: Explore potential differences in the acceptability of the adapted PEERS program between the first cohort and second cohort based on feedback from autistic adults and their study partners.
[Reviewer] The objectives are formulated imprecisely. They require a higher level of concreteness.
The wording of the (secondary) objective should refer not to the cohorts, but to the criteria that make them up.
There are two groups, one with ages between 25 and 55 (cohort 1) and another with ages between 21 and 68 years. This is not a differentiating criterion between both cohorts.
[Author] Method
Two types of data are identified to analyze:
a) Likert-type questionnaire, performing statistical analysis by comparing means using independent samples t-test
[Reviewer] a) In the case of the quantitative analysis proposal, the number of participants is very small, so the results do not provide more clarity than the mere presentation of the data descriptive of the results could. Furthermore, no personal variables are identified that allow us to consider that there are differences in relation to a specific characteristic that defines the groups to be compared.
[Author] b) Interview with qualitative data
[Reviewer] b) In the case of qualitative data there is no reference to how and what will be analyzed.
[Author] Results
Comparison of Cohorts 1 vs. 2 Quantitative Survey Results
there was a large effect size between Cohort 1 mid-point survey compared to Cohort 2 mid-point survey , with Cohort 2 rating the PEERS components higher (d=0.96), and the differences approached statistical significance [t(15)=1.87, p=0.08].
there was a medium effect size on the final survey with Cohort 2 rating the PEERS components higher than Cohort 1 (d=0.61), but Item was not statistics significant [ t( 15)=1.18, p=0.25].
Lastly , there was also a medium effect size on the final overall satisfaction with the program , with Cohort 2 rating the program higher than cohort 1 (d=0.59), but Item was not statistics significant [t(15)=1.16, p=0.26].
[Reviewer] The results presented are very few.
When interpreting the p value (2-tailed), the value is .08, meaning that the probability of obtaining a sample in which both groups differ more than the groups used is 8%. Since the significance level was set at 5%, it is less than 8%. Therefore, it is assumed that there are no significant differences between the two samples.
The number of participants may not be sufficient to assume sample representativeness.
There is no presentation of results of the qualitative information that is referenced.
[Author] Discussion
[Reviewer] The discussion section should be a discussion of the results presented.
It is difficult to justify that the statements made in this section are derived from the results presented.
Reviewer 4 Report
Comments and Suggestions for Authors
Specific suggestions:
Line 77 - change “intellectually” to “intellectual”
Line 126 – did you mean “or”? “And” implies that your participants each met all 3 of your inclusion criteria. I'd be surprised if everyone had ADOS-2, KBIT-2, and SRS-2 all completed.
Table 1 and 4, and line 122 – what is “M”. Mean? Needs to be defined at least somewhere in your paper.
Line 160 – you need to keep the “and” that had been struck out. There is no such thing as a quantitative open-ended question.
Line 258 – add “the” in front of “participant’s” to read smoother.
General comments:
Excellent work! This study has potential for high impact, given its potential applicability to autistic adults across the lifespan (other than the exceptions you've listed). As you've explained, PEERS is well regarded for young adults, but it's time for something similar and effective to be available for autistic adults from other age groups. I particularly like how your study was iterative, involving two consecutive cohorts. All programs like this should be iterative for continuous quality improvement.
Introduction:
Appropriately comprehensive in providing a background to your study and why it is important, including a description of the original PEERS.
Materials and Methods:
Great job including autistic adult input in multiple ways.
Tables were clearly presented. The role play scripts are excellent to allow reader to understand changes made between “risky” and “PEERS strategies”
Results:
Very clearly presented. I appreciate the changes made to original plans given many autistic adults did not have someone in their life to be a “mandatory” program partner.
Discussion:.
I agree with all your points. You did great work in adapting PEERS for other age groups and lessening ableist language / components of the original PEERS to reflect more acceptance of neurodiversity.
I agree with future suggestions as they accurately and comprehensively address the limitations of this study.
Round 2
Reviewer 2 Report
Comments and Suggestions for Authors
The current revision does not meet the expected standards, therefore, a more extensive correction is required to present the study in a better way. It is observed that the proposed adjustments are generally insufficient and the topics addressed need to be examined more in-depth. In addition, providing more detailed explanations of the analysis methodology and the results section could enhance the understandability and scientific contribution of the study. Specifically, it is recommended to broaden the participant base of the study and investigate the effects on individuals with various demographic features. Such a revision will enable the study to be presented in a stronger and more effective manner.
Comments on the Quality of English LanguageMinor editing of English language required
Reviewer 3 Report
Comments and Suggestions for Authors
The work still has methodological problems.
However, the topic is of great interest and its treatment is original.
The paper has gained in clarity and I believe it may be of interest to Healthcare readers.
These are the reasons for proposing its publication, although the article has methodological mismatches with respect to what is expected for a good research article.
